# The Daily Fire Hazard Index: A Fire Danger Rating Method for Mediterranean Areas

**Giovanni Laneve** [†,‡] , **Valerio Pampanoni** *[,†,‡] and **Riyaaz Uddien Shaik** [†,‡]

Scuola di Ingegneria Aerospaziale, Sapienza University of Rome, 00138 Rome, Italy;
giovanni.laneve@uniroma1.it (G.L.); riyaaz.shaik@uniroma1.it (R.U.S.)

* Correspondence: valerio.pampanoni@uniroma1.it; Tel.: +39-06-4991-9781

† Current address: Earth Observation Satellite Images Applications Lab (EOSIAL), School of Aerospace Engineering (SIA), Sapienza University of Rome, Via Salaria 851, 00138 Rome, Italy.

‡ These authors contributed equally to this work.

**Abstract:** Mediterranean forests are gravely affected by wildfires, and despite the increased prevention effort of competent authorities in the past few decades, the yearly number of fires and the consequent damage has not decreased significantly. To this end, a number of dynamical methods have been developed in order to produce short-term hazard indices, such as the Fire Probability Index and the Fire Weather Index. The possibility to estimate the fire hazard is based on the observation that there is a relationship between the characteristics of the vegetation (i.e., the fuel), in terms of abundance and moisture content, and the probability of fire insurgence. The density, type, and moisture content of the vegetation are modeled using custom fuel maps, developed using the latest Corine Land Cover, and using a number of indices such as the NDVI (Normalized Difference Vegetation Index), Global Vegetation Moisture Index (GVMI), and the evapotranspiration, derived from daily satellite imagery. This paper shows how the algorithm for the calculation of the Fire Potential Index (FPI) was improved by taking into account the effect of wind speed, topography, and local solar illumination through a simple temperature correction, preserving the straightforward structure of the FPI algorithm. The results were validated on the Italian region of Sardinia using official wildfire records provided by the regional administration.

**Keywords:** wildfires; mediterranean; hazard; MODIS

## 1. Introduction

The problem of forest fires affects all European countries, with varying degrees of gravity and sometimes with wildly different frequency. In particular, Mediterranean countries tend to be the most affected, especially if the number of fires is scaled to their surface area. According to the data gathered by the the European Forest Fire Information System (EFFIS), and taking into account only the wildfires recorded in a four-year time span that goes from 2014 to 2018 in France, Greece, Italy, Portugal, and Spain, 1,717,369 hectares of land were burned as a result of 144,837 wildfires [1–4]. Despite an increased prevention effort and technological support for the decision makers, the average values of burnt areas and number of wildfires were barely affected.

The S2IGI project was created in this context, as a result of a joint effort of The School of Aerospace Engineering of La Sapienza University, the Biometeorology Institute (IBIMET) of the National Research Council of Italy (CNR), and the Sardinian-based company Nurjana Technologies, with the objective to create an integrated fire management information system. S2IGI aims to cover all three operative phases of firefighting, namely prevention, detection, and damage assessment, by providing state-of-the-art software applications and systems based on satellite technologies [5]. The Daily Fire Hazard Index

(DFHI), in particular, enables data driven decisions in the pre-event phase, allowing decision makers to allocate resources on the territory on the basis of a model that uses up-to-date meteorological and satellite data to monitor the current state of vegetation Figure 1.

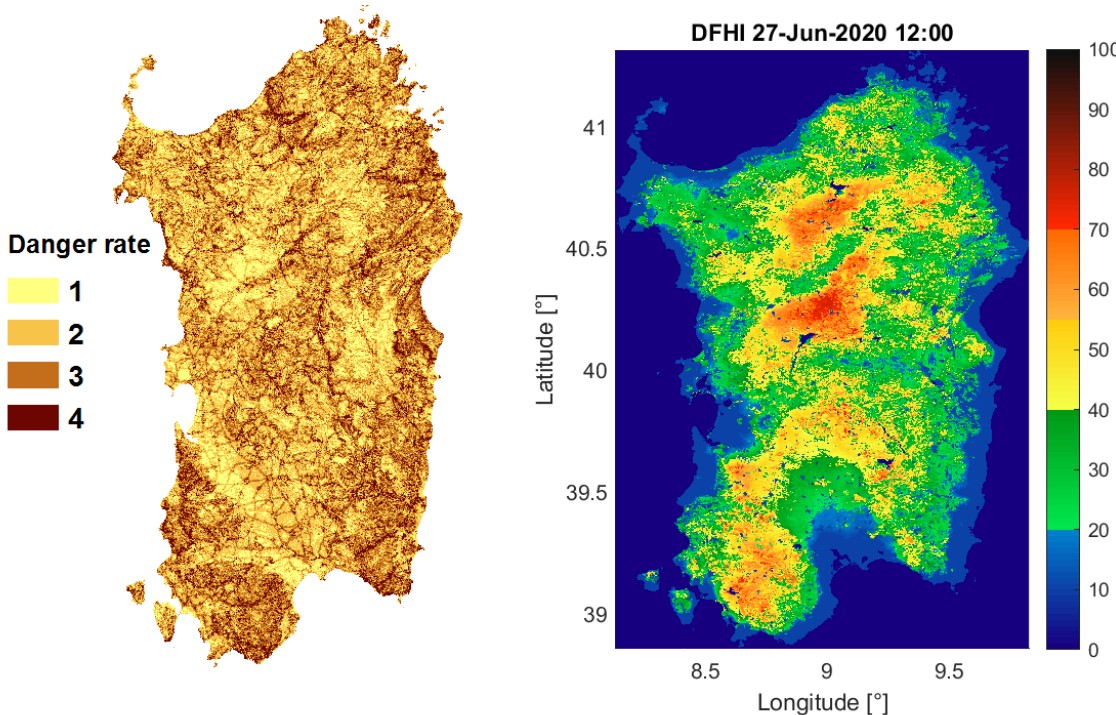

**Figure 1.** On the left, a map of static fire danger, an example of a static method of fire hazard estimation. On the right, a typical Daily Fire Hazard Index (DFHI) map, an example of a dynamical method for fire hazard estimation.

## *1.1. Methods of Fire Risk Estimate*

The idea to develop indices capable of predicting a fire hazard is based on the observation that there is a tight relationship between the fire and the characteristics of the fuel (vegetation type, density, humidity content), of the topography (slope, altitude, solar aspect angle), and the meteorological conditions (rainfall, wind direction and speed, air humidity, surface and air temperature). These parameters directly impact the proneness of fire ignition and the propagation of a given area. Since these quantities can be measured, either directly or remotely, the fire hazard can be considered strictly dependent from them, and can be estimated when such parameters are known, notwithstanding the fact that over 90% of ignitions in the Mediterranean area are caused by human actions, either intentional or accidental.

According to the adopted model, the methods of fire danger estimate exploit different information and can be distinguished into two main categories [6]:

- *Statistical* or *structural methods* (long-term fire risk indices), which define forecast models based on the utilization of slowly changing parameters like topography, vegetation type, or other variables that can be considered constant along the year, and statistical information on the frequency of the phenomenon;
- *Dynamical methods* (short-term fire risk indices) based on data measured continuously (i.e., daily), on the characteristics of territorial data (orography and vegetation), and on forecast models of meteorological parameters.

Typical hazard maps obtained using statistical and dynamical methods are shown in Figure 1.

### 1.2. State of the Art

Nowadays, given the increasing adoption of Earth Observation satellite data in the development of fire hazard indices and the concurrent decrease in popularity of statistical methods in favor of dynamic fire danger methods, it is more interesting to discriminate fire hazard indices according to whether they employ satellite data or not. Among the wide range of fire danger indices available worldwide, we can name a selection of those based on meteorological data [7], such as the Canadian Fire Weather Index System (CFWIS) and its components [8], the US National Fire Danger Rating System (NFDRS) and its components [9,10] and many single indices such as the Russian Nesterov Index [11] or the Italian RISICO (RISchio Incendi e Coordinamento) Index [12]. These indices all use standard meteorological data from open field stations or weather forecast models (as those provided by the ECMWF) (European Centre for Medium-Range Weather Forecasts) in order to assess the fire potential on a given day of the year.

The Australian fire danger rating is determined primarily [13] on the basis of McArthur's models for grasslands [14] and eucalyptus forests [15]. The ratings consist of categories of the Forest Fire Danger Index (FFDI) computed using the following inputs: Air temperature (T), relative humidity (RH), wind speed at 10 m height (V), a drought index (DI), which is a measure of soil moisture (based on the work of Keetch & Byram in the United States [16]), and a drought factor, which is a variable with values from 1 to 10 that represents the influence of recent temperatures and rainfall events on fuel availability.

The inputs for the Grassland Fire Danger Index (GFDI) are: Grassland curing (proportion of dead grass), relative humidity, air temperature, fuel weight, and wind speed. The equations for these systems can be found in [17].

The Canadian Forest Fire Danger Rating System [18] (CFFDRS) is determined across Canada each day of the fire season. The CFFDRS contains two major subsystems: The Canadian Forest Fire Weather Index (FWI) System and the Canadian Forest Fire Behavior Prediction (FBP) System. The FWI System [8] provides a means of evaluating the severity of fire based solely on weather readings. The resulting fuel moisture codes and fire behavior indices are based on a single standard fuel type that can be described as a generalized pine forest, mostly jack pine and lodgepole pine. The FBP System [19] relies on outputs from the FWI System and other site-specific information (such as topography and fuel type), and provides a quantitative assessment of fire behavior in terms of potential Surface Fuel Consumption, Rate of Spread, Total Fuel Consumption, Head Fire Intensity, and Crown Fraction Burned.

The inputs to the NFDRS, the current version of the U.S. National Fire Danger Rating System, include both weather and non-weather data like fuel types and topography. The NFDRS uses two types of weather data: One is an instantaneous description of weather (1300 LST observation), while the other is composed of summary statistics of weather over a 24-hour period, namely maxima, minima, and total.

In Europe, the JRC (Joint Research Center) carried out a comparative study of a number of selected national fire danger indices in the framework of the EFFIS program [20]. In 2007, it decided to focus research and development efforts on the components of the Canadian Fire Weather Index system. The list of different indices (six of which represent an evolution of indices developed for national applications, which are basically meteorological indices) compared by the JRC includes: The Portuguese index INMG [21], the ICONA Method [22], the Numerical Risk by Drouet and Sol [23], the Italian Index of Risk [24], the Canadian Fire Weather Index, the BEHAVE Model [25] and the Fire Potential Index (FPI) [26,27]. After this study, the JRC developed a common web-based service providing daily values of the FWI computed on a 10 km × 10 km grid.

Similar studies have been carried out in other regions using other methods, with the result that the FWI performed better than other methods in almost all instances. Consequently, the CFFDRS has become a common language not only among scientists but also between practitioners dealing with fire danger assessment. This kind of information may be used for prevention and management purposes,

such as the scheduling of patrols and aerial observations, public warnings, and fire-fighting resource allocation.

In Italy, the Civil Protection adopted the RISICO model, which is a fire danger rating system that was developed specifically for the vegetation cover of the Mediterranean [28]. RISICO integrates meteorological observations and forecasts from a NWP (Numerical Weather Predication) Limited Area Model (LAM) and ECMWF-Integrated Forecasting System (IFS) with vegetation cover and topography data. In addition to the weather, different fire behaviors with different fuel types and the effect of the local slope are taken into account in order to forecast fire spread. It is based on similar principles as the Canadian Fire Weather Index (FWI) in GEFF (Global ECMWF Fire Forecasting), but with a more detailed spatial, temporal scale and process description. The experience acquired during the development of RISICO also revealed the importance of the persistence of moisture content in fine dead fuels during an interval of 24–48 h [29].

### 1.3. The Role of Remote Sensing in Fire Hazard Assessment

Remote sensing provides a means for assessing vegetation status and monitoring changes over large geographic extents, making it a useful tool for supporting the development of fire hazard indices. Remote sensing systems allows one to collect biophysical measurements of ground conditions before and after fire events. These measurements have been used [30] in fire risk mapping [31–33], fuel mapping [34], active fire detection [35–39], burnt area estimates [40,41], burn severity assessing [42–44], and vegetation recovery monitoring [45]. Therefore it is not surprising that, in addition to the fully meteorological-based methods recalled above, several fire hazard estimation methods based exclusively on satellite data have been proposed. One of the most recent efforts in this sense [46] proposes a new Midterm Fire Danger Index (MFDI) for the Euro-Mediterranean region (Greece). Based on Earth Observation and ancillary geographic data, it provides reliable estimates of the fire ignition danger for a period of eight days without using any meteorological data. The same paper cites other techniques that are based on satellite-retrieved vegetation indices, land cover, surface temperature, and live fuel moisture.

It is worthwhile to note that throughout the present paper we will use the terms "fire hazard" or "fire danger" as synonyms, in order to express the proneness of an area to fire ignition or spread. Despite the effort made in [30], there exists no shared and universally accepted definitions of these terms in the fire management literature, and therefore this remark is necessary in order to specify the meaning that we assign to those terms to avoid any misunderstanding.

The fire hazard estimation methodology that we are presenting belongs to the family of methods, which incorporates both remote sensing and meteorological variables. In other words, in our case fire hazard maps based on satellite data combine meteorological forecast data, vegetation status information, vegetation fuel maps, and further ancillary information to predict the fire danger spatial distribution on local, national, or regional areas according to the characteristics of input data. Few examples of this methodology exist and we shall recall some of them, such as the pioneering work carried out by Burgan et al. [27], which will be described below.

For instance, Qin et al. [47] quantified the forest fire risk in China on a national level by using the Fuel State Index (FSI), previously adopted by Chuvieco [31], and the Background Composite Index (BCI). The FSI was estimated using Moderate Resolution Imaging Spectroradiometer (MODIS) data, meteorological data, and other basic data on the distribution of fuel types on national territory. The FSI and BCI were used to calculate the Forest Fire Danger Index (FFDI), which is regarded as a quantitative indicator for national forest fire risk forecasting and forest fire risk rating, shifting from qualitative description to quantitative estimation.

Chuvieco et al. [48] propose a comprehensive fire risk assessment system, in which satellite data are used to estimate the live vegetation Fuel Moisture Content (FMC). Furthermore, Fiorucci et al. [49] introduced the widely used satellite-based index NDVI (Normalized Difference Vegetation Index) as a

means to estimate the vegetation Relative Greenness (RG, [27]), which they obtained from a temporal series of SPOT-VEGETATION images. A similar process is described in the following paragraphs.

In some cases satellite data, in particular active fire data, are used to identify the factors that are mainly affecting fire occurrence probability [50], allowing the development of a fire hazard geospatial map or the calibration of fire danger models [51]. Considering that, except under extreme weather conditions, a fire almost never occurs (at least in Europe) without human intervention [52], accounting for the human factor is always mandatory in the definition of a fire hazard [53]. This is often achieved with static (long-term) Fire Hazard Maps, which usually take into account the statistical spatial distribution of fires in the previous five to ten years [31,53,54]. Nevertheless, a statistical analysis based on previous fire seasons cannot possibly consider the weather conditions of the present season, which can differ significantly from one year to another, or short-term (annual) changes in human behavior. In this case satellite data, and in particular real-time detection of hot spots can be used to register any change in human behavior. This has led to the development of the "seasonal" hazard index described in [52].

As mentioned before, pioneering work in the field of remote sensing for fire hazard mapping is represented by the work carried out by Burgan et al. [27], who proposed a model using both static and dynamic variables from three data sources: Fuel type maps, satellite sensor images, and meteorological data to derive an index referred to as Fire Potential. Starting from such a model,the JRC developed their own approach differing from the original one in terms of data sources, reference data for testing, and geographical area in which the model is applied. This fire hazard index is referred to as the Fire Potential Index (FPI), in accordance with the underlying model.

In the framework of the SIGRI (Sistema Integrato per la Gestione del Rischio Incendi) project, funded by the Italian Space Agency (ASI), we decided to continue the development of the FPI starting from the attempt made by the JRC to adapt an index originally developed for the United States to the conditions in the Euro-Mediterranean region [55]. The FPI consists in the estimate of fuel conditions by means of a separation of dead and green vegetation. This estimate is carried out by using maps of the vegetation index NDVI (Normalized Difference Vegetation Index) obtained with space-borne sensors such as, for instance, MODIS (on board of the Terra and Aqua satellites). To carry out the computation, a map of the fuel distribution on the area of interest is needed. This index, commonly called "integrated" or "advanced", is based on the FPI derived by Burgan [27] for the United States and successfully validated in California.

*1.4. The Daily Fire Hazard Index*

The JRC, after a campaign of comparative studies among different fire hazard indices (carried out in the framework of the EFFIS initiative), decided to adopt the FWI to provide fire danger maps for a European level [56] despite the fact that it does not employ satellite data. For this reason, we decided to further develop the FPI by adding useful information in order to improve its capability to capture the spatial distribution of the hazard. We called our index the DFHI (Daily Fire Hazard Index). As better described in the following paragraphs, the improvements involve:

- The estimate of the moisture content in live vegetation;
- The introduction of the local solar aspect effect;
- The introduction of the effect of the wind in increasing the vegetation proneness to burn.

The importance of wind effect in determining fire hazard conditions is well known in the Sardinia region, which is the area of interest of this study. The model requires the NDVI to compute the Relative Greenness, meteorological data (air temperature and relative humidity) for estimating the Ten-Hours Timelag Fuel Moisture, (FM10hr), and a fuel map to estimate the percentage of dead vegetation. The Relative Greenness (RG) or vegetation stress index represents how green a pixel is, with reference to the range of historical observation of the NDVI selected for the analysis [26]. This quantity enables the estimate of the percentage of green fuel as a function of the fuel model assigned to each pixel.

The Ten-Hours Timelag Fuel Moisture, has been selected as the best quantitative representation of the humidity retained by the dead vegetation [57]. Such a quantity can be computed by using the meteorological parameters and the relationship described by Lopez [55]. The computation of the new version of our DFHI (Daily Fire Hazard Index), which takes into account the JRC experience [55], the development carried out in the framework of the SIGRI project [58] and the further improvement achieved in the framework of the PREFER project [52,59] will be described in the following paragraphs.

## 2. The DFHI Algorithm

A schematic of a simplified version of the DFHI algorithm can be found in Figure 2. In the following sections we will describe thoroughly the steps followed to compute the index starting from the satellite images and the meteorological data.

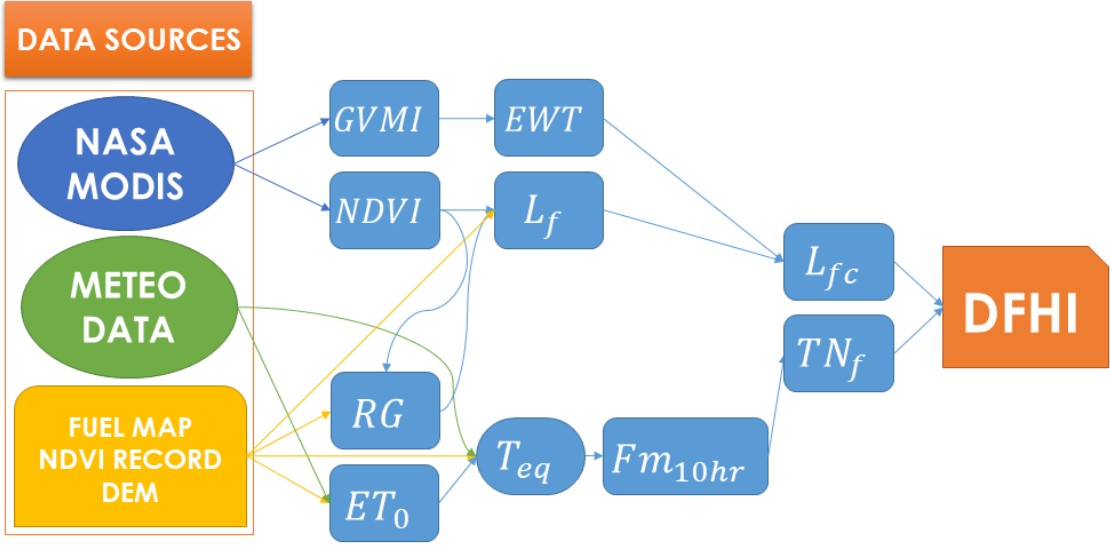

**Figure 2.** Simplified flowchart of the DFHI (Daily Fire Hazard Index) algorithm.

### 2.1. Data Sources

The data sources of the algorithm are summed up in Table 1.

**Table 1.** Source Data: April 2020 Version.

| Dataset | Data Type | Resolution [m] | Provider | Date/Frequency |
|---|---|---|---|---|
| SAR[1]-derived DEM[2] | Raster | 10 | Regione Sardegna | 2011 |
| NDVI[3] Record | Raster | 250 | - | 2014–2019 |
| Fuel Map | Shapefile | - | JRC[4] | 2013 |
| MOD09GA Images | Raster | 500 | NASA LAADS DAAC | Daily |
| MOD09GQ Images | Raster | 250 | NASA LAADS DAAC | Daily |
| Meteo Data | Raster | 3000 | WRF[5] | Daily |

[1] Synthetic Aperture Radar, [2] Digital Elevation Model, [3] Normalized Differential Vegetation Index, [4] Joint Research Center, [5] Weather Research and Forecasting model

The MODIS09GQ and MODIS09GA products contain daily surface reflectances at a 250 and 500 m resolution respectively, and the datasets used for our purposes are described in Table 2.

**Table 2.** MODIS Products Used in the DFHI Algorithm.

| Product | HDF Layer | Description | Resolution [m] | Wavelength [nm] |
|---------|-----------|-------------|----------------|-----------------|
| MOD09GQ | sur_refl_b01_1 | Surface Reflectance Band 1 | 250 | 620–690 |
| MOD09GQ | sur_refl_b02_1 | Surface Reflectance Band 2 | 250 | 841–876 |
| MOD09GA | sur_refl_b02_1 | Surface Reflectance Band 2 | 500 | 841–876 |
| MOD09GA | sur_refl_b06_1 | Surface Reflectance Band 6 | 500 | 1628–1652 |

In addition to the daily updated satellite and meteorological data, the algorithm employs a Digital Elevation Model (DEM), a five-year NDVI record of the area of interest, and a fuel map. The Digital Elevation Model is provided by the Sardinian administration, and was upscaled to 250 m in order to match the resolution of the MOD09GQ images, which is also the output resolution of the DFHI map. The NDVI record was built at La Sapienza using MOD09GA images, and the fuel map was compiled by the JRC and provided to us in the context of the PREFER project.

## 2.2. The DFHI Equation and the Hazard Classes

The final expression used to determine the DFHI is the following [55] :

$$DFHI = (1 - L_f)(1 - TN_f) \cdot 100 \qquad (1)$$

where:

- $L_f$ fraction of live vegetation (dimensionless);
- $TN_f$ fraction of ten-hours timelag fuel moisture (dimensionless).

Ultimately, it is through these two metrics that we assess the fire hazard. On one hand, the ones' complement of the fraction of live vegetation $L_f$, that is to say the fraction of dead vegetation, is a measure of the quantity of fuel available for the wildfires. On the other hand, the ones' complement of the ten-hours timelag fuel moisture is a measure of the dryness of the fuel available for the wildfires. These are the two main aspects that must be modeled in order to provide a realistic hazard assessment and the DFHI does so by employing the latest meteorological and satellite data, in addition to fuel maps and historical records of the area of interest. Finally, the multiplication by the factor 100 allows us to obtain an easily readable index that holds values in the $[0, 100]$ interval, and that can be translated to a categorical variable in terms of hazard classes as described in Table 3.

**Table 3.** DFHI Hazard Classes.

| DFHI Interval | Hazard Class |
|---------------|--------------|
| 0–20 | No Hazard |
| 20–40 | Low Hazard |
| 40–55 | Medium Hazard |
| 55–70 | High Hazard |
| 70–100 | Very High Hazard |

In the following paragraphs we will describe in detail the procedure followed to generate daily DFHI maps.

## 2.3. Estimation of the Live Vegetation Fraction

In order to determine the likelihood of wildfire insurgence as accurately as possible, the DFHI relies on a number of metrics to distinguish live and dead vegetation, and subsequently to estimate its moisture content. This process allows us to quantify the available fuel for the fire on one hand, and to realistically assess its proneness to burn on the other.

Firstly, NRT MODIS imagery, in particular the MOD09GQ product, is used to compute the NDVI:

$$NDVI = \frac{NIR - RED}{NIR + RED} \tag{2}$$

where the MODIS surface reflectances $sur\_refl\_b02\_1$ and $sur\_refl\_b01\_1$ are used for the near-infrared and red channels respectively. The same reflectances are used to create a cloud mask in order to fill cloud pixels with a `NoData` value. What ultimately allows us to distinguish live and dead vegetation is the comparison between the currently measured NDVI and its historical record for a given pixel, through the metric Relative Greenness, which is defined as follows:

$$RG = \frac{NDVI - min_{5y}NDVI}{max_{5y}NDVI - min_{5y}NDVI} \cdot 100 \tag{3}$$

where $min_{5y}NDVI$ and $max_{5y}NDVI$ are the minimum and maximum NDVI values registered in a given pixel in the last five years. Similar to the NDVI, the RG is a dimensionless quantity, and its fractional value $RG_f$ can be obtaining by omitting the 100 factor in Equation (3). The NDVI and the fractional value of the RG are then used to determine the fraction of green vegetation $L_f$, which is a corrected version of the $RG_f$ that accounts for the highest historical values of the NDVI recorded in a given pixel in the selected time period:

$$L_f = \frac{RG_f \cdot \left[ 35 + 40 \cdot \frac{NDVI_{mx} - 100}{80} \right]}{100}. \tag{4}$$

The $NDVI_{mx}$ variable that allows us to rescale the RG in relation to the NDVI records of a given pixel is calculated as follows:

$$NDVI_{mx} = 100 \cdot max_{5y}NDVI + 100. \tag{5}$$

Subsequently, channels 2 (NIR) and 6 (SWIR) of the 500 m resolution MOD09GA product are downscaled to 250 m and used to compute the Equivalent Water Thickness (EWT), which is our fundamental indicator of the moisture content of vegetation. The seasonal decrease of this metric has been directly linked to the start of the wildfire season [60], and in the form described by [61], it can be determined using the Global Vegetation Moisture Index (GVMI) as a precursor.

$$GVMI = \frac{(NIR + 0.1) - (SWIR + 0.02)}{(NIR + 0.1) + (SWIR + 0.02)} \tag{6}$$

Using Ceccato's form, we can therefore use the MOD09GA NRT reflectances $sur\_refl\_b02\_1$ and $sur\_refl\_b06\_1$ to derive daily maps of the EWT:

$$EWT = \frac{-(ad + c - d \cdot GVMI) + \sqrt{(ad + c - d \cdot GVMI)^2 - 4cd(a + b - GVMI)}}{2dc}. \tag{7}$$

The coefficients $a, b, c$ and $d$ were adapted from those provided by Ceccato [61] for SPOT/VEGETATION imagery in order to correctly apply Equation (7) to our MODIS images ($a = 0.4743, b = -0.3967, c = 5.853e - 5, d = 0.006577$). The EWT is then normalized using its mean $\langle EWT \rangle$ and standard deviation $\sigma(EWT)$ computed on the current area of interest:

$$\overline{EWT} = \frac{EWT}{\langle EWT \rangle + 2\sigma(EWT)}. \tag{8}$$

This normalized version of the EWT is used to correct the fraction of green vegetation $L_{fc}$:

$$L_{fc} = L_f \left[ 1 + 0.2(\overline{EWT} - 1) \right]. \tag{9}$$

This correction allows us to model the fact that an increase in EWT results in a lower fire hazard. The lower fire danger associated to a higher EWT is simulated as an increase in fraction of live vegetation.

*2.4. Modeling of the Effect of the Wind through the Evapotranspiration*

Daily meteorological forecast data of temperature, humidity, and wind speed are used to estimate the moisture content of the so-called "small dead fuels", defined as those that take ten hours to lose 63% of the difference between their initial moisture content and the equilibrium moisture content in an atmosphere at constant temperature and humidity. These fuels are usually referred to as Ten Hour Lag Fuels, and their moisture content can be estimated through the metric Ten-Hours Timelag Fuel Moisture, which is denoted as $Fm_{10hr}$, expressed as a percent moisture content and calculated as follows [55]:

$$Fm_{10hr} = 1.28 \cdot emc \qquad (10)$$

where the variable *emc* stands for equivalent moisture content, which is a function of the air moisture and temperature at two meters of height. If the *emc* and the relative humidity are expressed as a percent moisture and humidity content and the temperature is expressed in Fahrenheit degrees, this metric can be calculated using the empirical relationship described in Fosberg's June 1977 USDA Forest Research Paper [62], which constitutes one of the most important theoretical foundations of vegetation modeling. If the temperature is expressed in Celsius degrees, the *emc* can be calculated as follows:

$$\begin{cases} emc = 0.03229 + 0.262573\,h - 0.001\,hT & \text{if } h < 10\% \\ emc = 1.7544 + 0.160107\,h - 0.02661\,T & \text{if } 10\% \leq h \leq 50\% \\ emc = 21.0606 + 0.005565\,h^2 - 0.00063\,hT - 0.494399\,h & \text{if } h > 50\% \end{cases}$$

The original American procedure uses only data gathered by meteorological stations and subsequently interpolated using an inverse distance squared algorithm to calculate this quantity. Lopez [55] uses a slightly different procedure that involves a high number of meteorological stations and a more complex interpolation algorithm, and furthermore states that the humidity and temperature values are corrected for solar heating. Nevertheless, the methodology used to perform the correction is not explained.

Our procedure improves upon the scientific literature by taking into account the effect of wind speed in a very straightforward way, by making use of the reference evapotranspiration determined using the Penman–Monteith equation and following the Food and Agriculture Organization (FAO) guidelines [63], which can be consulted for an extensive and detailed explanation on its calculation procedure. For the sake of simplicity and brevity, we will only show the final expression of the evapotranspiration and explain how we used it to account for the effect of the wind speed on the state of the vegetation.

The reference evapotranspiration, denoted as $ET_0$, can be calculated using the following expression shown in the FAO Irrigation and Drainage Paper 56 [63]:

$$ET_0 = \frac{0.408\Delta(R_n - G) + \gamma\frac{900}{T+273}u_2(e_s - e_a)}{\Delta + \gamma(1 + 0.34u_2)} \qquad (11)$$

where:

- $ET_0$ reference evapotranspiration [mm day$^{-1}$];
- $R_n$ net radiation at the crop surface [MJ m$^{-2}$ day$^{-1}$];
- $G$ soil heat flux density [MJ m$^{-2}$ day$^{-1}$];
- $T$ mean daily air temperature at 2 m height [°C];
- $u_2$ wind speed at 2 m height [m s$^{-1}$];
- $e_s$ saturation vapour pressure [kPa];
- $e_a$ actual vapour pressure [kPa]

- $e_s - e_a$ saturation vapour pressure deficit [kPa];
- $\Delta$ slope vapour pressure curve [kPa $°C^{-1}$];
- $\gamma$ psychrometric constant [kPa $°C^{-1}$].

As specified in the description of each parameter, all meteorological variables should be either measured or converted at 2 m of height to maintain uniformity, and in general the measurements should be taken above an extensive surface of green grass that covers the soil and is not short of water. In our case, the reference evapotranspiration is calculated for all the pixels in the area of interest, using the latest available meteorological data at 2 m height and in particular the temperature and wind speed. Concurrently, a look-up table of $ET_0$ values is created using the same inputs except for the topography, the wind speed,—which is kept constant at 2 m/s—and the temperature, which is varied in an interval of temperatures $T^* \in \{0, 2, 4, \ldots, 64\,°C\}$. The measured evapotranspiration of each pixel is then compared with the values contained in the look-up table and the temperature $T_{eq}$ is assumed to be the one that corresponds to the minimum evapotranspiration difference.

$$T_{eq} := T^* : |ET_0(T^*) - ET_0(T)| = min_{T^* \in \{0,2,\ldots,64°C\}} |ET_0(T^*) - ET_0(T)| \tag{12}$$

If the difference between the equivalent temperature and original air temperature at a 2 m height is smaller than $1\,°C$, no correction is applied. Otherwise, the equivalent temperature $T_{eq}$ is assigned to the pixel except in two specific cases:

- If the wind speed is lower than 20 m/s and the temperature difference is higher than 15 °C, only a 15 °C correction is applied. This is done in order to account for the fact that, even though evapotranspiration tends to increase the temperature of the vegetation, air always tends to circulate, and therefore a larger correction would be unrealistic;
- If the temperature difference is lower than $-15\,°C$, only a $-15\,°C$ correction is applied regardless of the wind speed.

Thus, the effect of the wind speed is modeled through a correction of the temperature of the vegetation using the evapotranspiration as a proxy. Furthermore, the Penman–Monteith equation also allows us to include the effect of solar illumination and of the local topography through a simple correction of the pixel temperature. The $Fm_{10h}$ calculated using the equivalent temperatures is then converted to its fractional value $TN_f$ using the following equation:

$$TN_f = \frac{Fm_{10h} - 2}{MX_d - 2} \tag{13}$$

where $MX_d$ represents the extinction moisture of the dead fuel, which is the moisture content that a certain type of dead fuel must have in order to prevent the fire from spreading [64]. The value of this quantity depends on the fuel type.

## 3. Results

In order to assess the performance of the DFHI in predicting the fire hazard, daily DFHI values were compared to the latest official wildfire records in the area of interest, which at the time of writing date back to the year 2017. DFHI maps were generated for the entire fire season of 2017, conventionally assumed to start the first of June and to end on 31 August. The area of interest, which consists of the region of Sardinia, is particularly suitable for our purposes, since it is well known that the most damaging wildfires occur in days of strong Mistral winds [65].

The DFHI maps for validation were generated using non-NRT MOD09GA and MOD09GQ L2 products for the reference day, and gridded meteorological data provided by the Italian Air Force. First of all, the DFHI values obtained for each day and each land pixel were converted, their respective hazard class are as described in Table 3, and recorded into a bar plot in order to visualize the distribution of the index over the entire area of interest without any constraint on the land cover

type. Subsequently, the polygons of the burnt areas contained in the 2017 fire season shapefile were rasterized into boolean images, where the `true` values was assigned to the burnt pixels and the `false` value to all the other pixels. For each day of the fire season, we recorded the DFHI classes calculated for each pixel of the burnt areas, provided that the total burnt area was higher than a threshold value of 100,000 m$^2$ = 10 ha. This specific threshold was chosen for two main reasons: Firstly, in Sardinia, as in many other Mediterranean regions, most of the fires are caused by humans, and therefore a useful fire hazard index should identify the areas where the vegetation status is more prone to sustain a wildfire over large areas, and it would therefore be pointless to include small, short-lived fires in our analysis and secondly setting the threshold at this value still allowed us to include a relatively high number of fires and carry out significant statistical analysis.

Finally, the global DFHI distribution and the DFHI distribution restricted to the burnt areas were represented using side-by-side bar plots. This allows us to compare the global distribution of the DFHI with its distribution over burnt areas at a glance, and to assess its performance and to quickly identify any bias, either towards higher or lower hazard classes. This process was carried out firstly using the old DFHI algorithm that did not correct the temperature using the evapotranspiration, and the results are shown in Figure 3. Secondly, the process was repeated using the latest version of the DFHI algorithm discussed in this article, whose results are shown in Figure 4. Note that the entire terrain of the region of Sardinia is included in this analysis, without excluding a priori land cover types that are not usually a target for wildfires, such as urban areas or sandy areas.

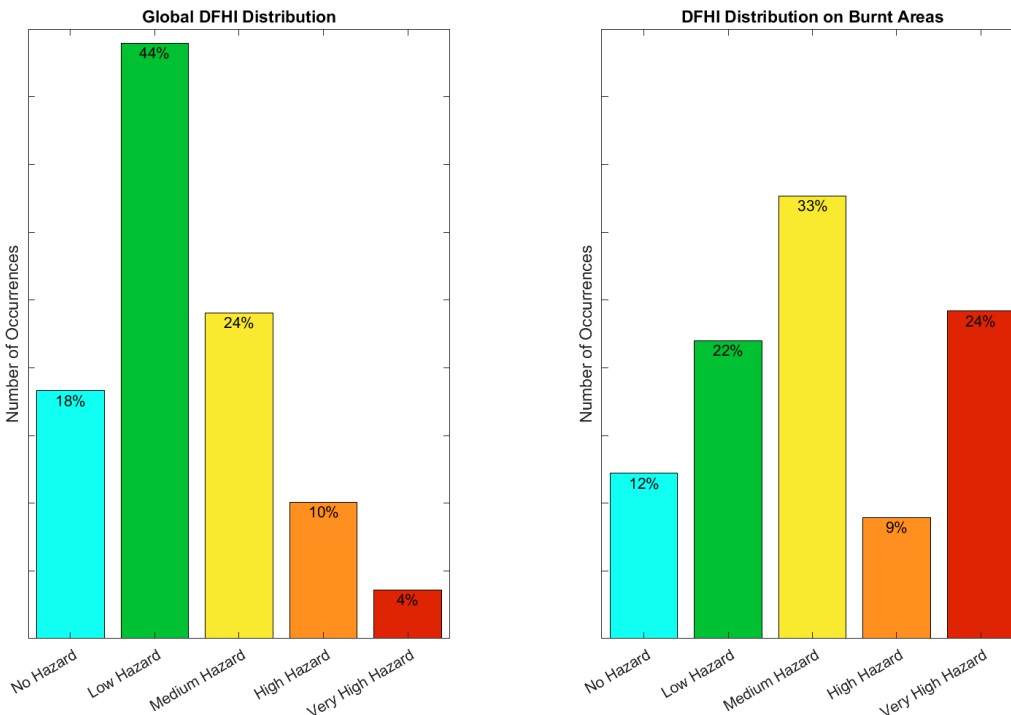

**Figure 3.** Distribution of the DFHI on the entire Area Of Interest (AOI) (**left**) and only on burnt areas (**right**) without applying temperature correction.

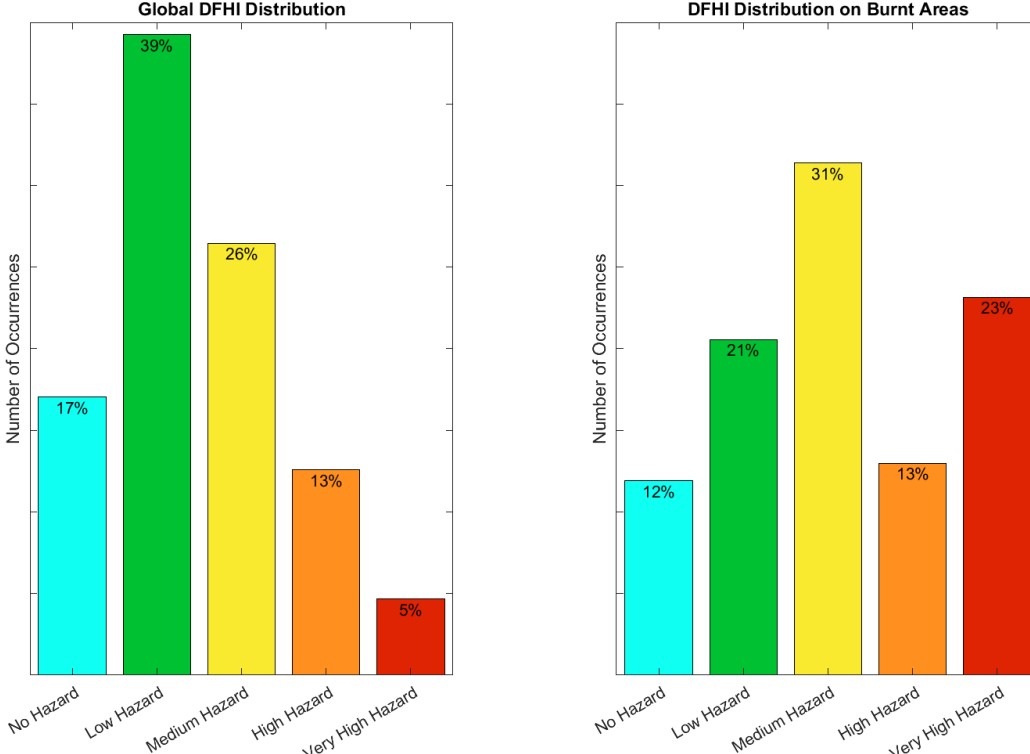

**Figure 4.** Distribution of the DFHI on the entire AOI (**left**) and only on burnt areas (**right**) after applying temperature correction.

## 4. Discussion

The global DFHI distribution was positively skewed, and shows that the overwhelming majority of the pixels of the area of interest, roughly 86% of the total count, fell into the three lower hazard categories, while the higher hazard classes combined amount for the remained 14%. In particular, only 10.08% of the pixels fell into the "High Hazard" class, and an even lower 3.61% fell into the "Very High Hazard" class, implying that the index was not inherently biased towards the highest hazard classes. If we compare this distribution with the one obtained on the burnt areas, the results are noticeably different: Roughly one third of the pixels that were reportedly burned by wildfires in the year 2017 were classified as "High Hazard" or "Very High Hazard" on the day they burnt, and in particular, a quarter of them was classified as "Very High Hazard".

Slightly better, but qualitatively analogous results were obtained with the latest version of the algorithm. The global distribution was still positively skewed, but this time the three lower hazard classes account for slightly less than 83% of the total pixel count, while the highest hazard classes account for the remaining 17%, against the 13% of the previous version. The same considerations apply to the distribution on the burnt areas: In general, the new algorithm tended to return more "High" and "Very High Hazard" pixels. This tendency was to be expected given the fact that the evapotranspiration tends to increase the temperature of the vegetation, and therefore to increase the fire hazard during the wildfire season, confirming the usefulness of the methodological improvements which is the object of this paper.

Unfortunately, the number of wildfires that satisfied the condition on the minimum burnt area in 2017, while significant, was not sufficient to draw definitive conclusions. We are currently in the process of building a software infrastructure capable of obtaining meteorological data on-demand, which will allow us to complete the validation using all the wildfire records available for the area of interest, which date back to 2005.

*Conclusions and Future Developments*

The latest version of the DFHI algorithm improved upon its predecessors with the inclusion of the effect of the wind speed, the topography of the area of interest, and the solar illumination conditions on the state of the vegetation by exploiting the Penman–Monteith equation and the FAO guidelines for the reference evapotranspiration. More generally, it provided a useful alternative to the other short-term, medium-resolution hazard indices because it made use of both the latest satellite data and meteorological forecast data to assess fire hazards on a daily basis, and could therefore be used by the decision makers to take countermeasures based on up-to-date remotely sensed data. The index does not show any bias towards lower or higher risk classes, and predicts the areas at risk over the area of interest with good reliability.

The possibility of creating high or very high resolution DFHI maps on protected areas of great naturalistic or historical value is being explored. High resolution DFHI maps could also be useful along the so-called Wildland Urban Interfaces (WUI), i.e. vegetated areas on the boundaries of urban settlements and infrastructures. Since almost all wildfires are started, either voluntarily or not, by humans, having high resolution hazard maps on these areas could greatly improve the decision makers' ability to identify higher risk areas. Furthermore, we are evaluating the possibility of using hyperspectral data provided by the PRISMA mission of the Italian Space Agency to upgrade the current fuel map. The DFHI validation will be repeated after the exclusion of these areas, and we expect to obtain even better results on the performance of the algorithm.

**Author Contributions:** Conceptualization, G.L.; Methodology, G.L.; Software, V.P. and R.U.S.; Validation, V.P.; Writing—original draft, V.P.; Writing—review & editing, G.L. and R.U.S. All authors have read and agreed to the published version of the manuscript.

**Funding:** The S2IGI project was funded by the Sardinian Regional Administration in the framework of the European Regional Development Fund, POR-FESR 2014-2020 initiative for intelligent and sustainable development, Axis 1 Action 1.2.2.

**Conflicts of Interest:** The authors declare no conflict of interest. The funders had no role in the design of the study; in the collection, analyses, or interpretation of data; in the writing of the manuscript, or in the decision to publish the results.

## Abbreviations

The following abbreviations are used in this manuscript:

| | |
|---|---|
| AOI | Area of Interest |
| ASI | Agenzia Spaziale Italiana |
| CFFDRS | Canadian Forest Fire Danger Rating System |
| CFWIS | Canadian Fire Weather Index System |
| CNR | Consiglio Nazionale delle Ricerche |
| DFHI | Daily Fire Hazard Index |
| ECMWF | European Centre for Medium-Range Weather Forecasts |
| EOSIAL | Earth Observation Satellite Images Applications Lab |
| EFFIS | European Forest Fire Information System |
| EWT | Equivalent Water Thickness |
| FAO | Food and Agriculture Organization |
| FFDI | Forest Fire Danger Index |
| FBP | Fire Behaviour Prediction |
| FPI | Fire Potential Index |
| FWI | Fire Weather Index |
| GEFF | Global ECMWF Fire Forecasting |
| GFDI | Grassland Fire Danger Index |
| GVMI | Global Vegetation Moisture Index |

IBIMET    Istituto di BIoMETeorologia
IFS       Integrated Forecasting System
JRC       Joint Research Center
LAM       Limited Area Model
MFDI      Midterm Fire Danger Index
NFDRS     National Fire Danger Rating System
NWP       Numerical Weather Prediction
NDVI      Normalized Difference Vegetation Index
PREFER    Space-based information support for the Prevention and Recovery of Forest Fires Emergency
          in the Mediterranean Area
RG        Relative Greenness
RISICO    RISchio Incendi e COordinamento
S2IGI     Sistema Satellitare Integrato di Gestione Incendi
SIA       Scuola di Ingegneria Aerospaziale
SIGRI     Sistema Integrato per la Gestione del Rischio Incendi
WUI       Wildland Urban Interface

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
