# Peer review of "The Daily Fire Hazard Index: A Fire Danger Rating Method for Mediterranean Areas"

_remotesensing, doi:10.3390/rs12152356_

Round 1

Reviewer 1 Report

The authors provide an updated and improved "daily fire hazard" index over the Mediterranean area of Sardinia.

The contribution is of interest, but improvements are required both in terms of style of presentation and the analysis per se.

Specific comments

  1. Please remove the word "tool" from the title
  2. A major issue that the authors must address relates to the consistency in the terminology related to fire risk, hazard, danger etc. Please define and cite accordingly
  3. The literature review is rather shallow, and does not include recent research efforts, especially within the Mediterranean areas (i.e. Stefanidou et al. 2019 etc). Section 2.1 should be expanded
  4. Line 166 and elsewhere. Please provide background info/details for all MODIS products used in the study
  5. Line 302. How did this fire size threshold was selected?
  6. Line 322. A statistical comparison among these 2 distributions is missing
  7. Section 4. Comparison within another area is strongly advised in order to prove the robustness and the strength of the new approach

Reviewer 2 Report

In this work, authors propose an improvement of a pre-existing fire index by taking into account extra variables focused on the state of vegetation and fine fuel, using MODIS imagery, fuel maps and WRF models. Results showed an improvement of this index, which is of interest to advance in fire prediction. Thus, I think this work is suitable for publication in Remote Sensing.

General comments:

Please, specify better in the abstract which are the improvements you performed over the old index. It could be misleading reading the abstract, where you explain that your corrections are about “the effect of wind speed, topography,and local solar illumination conditions on the state of the vegetation” and then reading the footnote of Figures 3 and 4, where your correction is called “temperature correction”.

The structure of the paper is atypical (1. Introduction, 2. State of the Art, 3 The DFHI algorithm, and 4. Validation of the results) in order to fit better the main purpose. Although acceptable, I wonder if the structure of the manuscript could be improved by joining 1. Introduction and 2. State of the Art sections.

The section Validation of results should be improved. Please, think about a better way to show the improvements of your index over the old one (Figs. 3 and 4). Some comments about this section are exposed below.

Line by line comments:

L100: Please, check if the expression is OK.

L107: It is necessary a reference for this statement.

L167: Why your NDVI obtained from MOD09GA has 250m spatial resolution if MOD09GA has 500m?

L293-294: This is methods, in your case, this should be explained in section 3 “The DFHI algorithm”.

L328-332 and L350-351: This paragraphs may go against your interests, and it raises the question on why you don’t wait to publish the 2005-2017 validation, or why you have not used a land-use mask which is not complicated.
